# Molecular Pathways of Altered Brain Development in Fetuses Exposed to Hypoxia

**DOI:** 10.3390/ijms241210401

**Published:** 2023-06-20

**Authors:** Anna Orzeł, Katarzyna Unrug-Bielawska, Dagmara Filipecka-Tyczka, Krzysztof Berbeka, Natalia Zeber-Lubecka, Małgorzata Zielińska, Anna Kajdy

**Affiliations:** 1Centre of Postgraduate Medical Education, I-st Department of Obstetrics and Gynecology, 01-813 Warsaw, Polanddagmara.filipecka-tyczka@cmkp.edu.pl (D.F.-T.);; 2Department of Genetics, Maria Sklodowska-Curie National Research Institute of Oncology, 00-001 Warsaw, Poland; 3Centre of Postgraduate Medical Education, Department of Gastroenterology, Hepatology and Clinical Oncology, 01-813 Warsaw, Poland

**Keywords:** perinatal hypoxia, fetus growth restriction, animal models

## Abstract

Perinatal hypoxia is a major cause of neurodevelopmental impairment and subsequent motor and cognitive dysfunctions; it is associated with fetal growth restriction and uteroplacental dysfunction during pregnancy. This review aims to present the current knowledge on brain development resulting from perinatal asphyxia, including the causes, symptoms, and means of predicting the degree of brain damage. Furthermore, this review discusses the specificity of brain development in the growth-restricted fetus and how it is replicated and studied in animal models. Finally, this review aims at identifying the least understood and missing molecular pathways of abnormal brain development, especially with respect to potential treatment intervention.

## 1. Introduction

Understanding, preventing, and treating abnormal fetal brain development due to prenatal hypoxia is one of the most significant challenges of perinatal medicine. The ability of clinicians to identify fetuses at risk of hypoxia has primarily increased due to a better understanding of abnormal placentation, the development of fetal growth restriction (FGR), and the use of ultrasound and computerized cardiotocography (cCTG). Unfortunately, these interventions only aim to prevent fetal death and fail to predict the degree of neurological impairment. For decades, neurological damage has been believed to result from injury during labor. However, modern pathophysiology has revealed that more than 80% of neurobehavioral problems in children have their origin from much earlier during pregnancy. Neurological damage can be overt, mild, or subtle depending on the degree and timing of hypoxia. To better understand brain development at the molecular level, animal models have been studied to identify pathways of altered brain development. Postnatal imaging of brain development using magnetic resonance imaging (MRI) has helped us to better understand which parts of the brain are most affected by hypoxia. Now, the performance of neurosonography allows for the identification of these changes prenatally. A complete understanding of this topic is needed to better understand how and when brain lesions occur. Furthermore, this could lead to possibilities of reversing the damage using therapies that promote brain development both in utero and postnatally.

This review aims to present the current knowledge on brain development resulting from hypoxia that is related to fetal growth restriction, including the causes, symptoms, and means of predicting the degree of brain damage. Furthermore, the review discusses the specificity of brain development in the growth-restricted fetus and how this is replicated and studied in animal models. Finally, this review aims to identify the least understood and missing molecular pathways of abnormal brain development, especially in the aspect of potential treatment interventions.

## 2. Causes of Prenatal Hypoxia

The emergence of intrauterine hypoxia depends on two main interdependent mechanisms: placental and environmental hypoxia. External factors influencing the proper oxygenation of the fetus involve maternal comorbidities, smoking, alcohol consumption, malnutrition, intrauterine infections, or maternal stress [1]. As an example, gestational diabetes mellitus leads to maternal and fetal metabolic dysfunction, which limits red blood cells production, alters organogenesis, and consequently causes placenta-mediated obstetric complications and hypoxia. As presented in the work of Perna et al., the neural consequences of gestational diabetes involves lower general intelligence, attention weaknesses, and problems controlling emotions [2,3]. The environmental causes of prenatal hypoxia are pre-placental causes of hypoxia and concern maternal and fetal health problems.

On the contrary, hypoxia due to placental dysfunction exclusively affects the fetus. On a molecular level, oxygen insufficiency causes damage to the mitochondrial electron transfer system (ETS), which promotes the formation of reactive oxygen species (ROS). ROS primarily affect the syncytiotrophoblast, which is susceptible to ROS superoxide (O2−) and hydrogen peroxide (H_2_O_2_) [4]. Low oxygen concentration along with oxidative stress leads to the release of damage-associated molecular patterns (DAMPs), including uric acid, high-mobility group box 1 (HMGB1), cell-free fetal DNA, S100 calcium binding proteins A8 and A12 (S100A8, S100A12), and heat shock protein 70 (HSP70). Simultaneously, cytokine levels are increased, especially pro-inflammatory cytokines (IL-1α, IL-1β, IL-6, IL-8) [5]. These changes are known as sterile inflammation, which mediates trophoblast necrosis and cell senescence [5,6]; this process is followed by a defective invasion of the trophoblast and abnormal remodeling of the maternal vasculature. Therefore, the histological picture of hypoxic damage of the placenta involves maternal malperfusion (MMP) and fetal vascular malperfusion (FVM) [7,8]. MMP lesions in the placenta are commonly associated with pregnancies that are complicated by FGR, preeclampsia, or both. Typical changes in the placenta include decidual vasculopathy (persistence of muscularized basal plate arteries, mural hypertrophy of membrane arterioles, and acute atherosis), villous or parenchymal infarctions, and laminar necrosis. An additional problem that was underlined in the work of Parks was decreased placental weight, which was also related to preeclampsia and FGR [7,9].

Induced epigenetic changes, such as oxygen-regulated gene expression, are mediated by hypoxia-inducible factor 1 complexes (HIFs) and other components of the HIF transcription complex that are activated in low oxygen concentration. Under normoxic conditions, HIF-1α is rapidly decomposed by the pVHL-mediated ubiquitin-proteasome pathway, while hypoxia blocks its degradation, leading to HIF-1α accumulation [4]; this induces the transcription of multiple genes that regulate adaptation and angiogenesis and promote anaerobic metabolism. Due to their pro-apoptotic effect, HIFs promote tumor progression and act as mediators in the development of cardiovascular and pulmonary diseases [4,10]. On the other hand, according to Yver et al., an absence of HIF-1α during gestation increased the developmental arrest and lethality. The main complications involved cardiovascular malformations, neural tube defects, and endothelial and blood vessel development [11].

## 3. Symptoms of Brain Damage Related to Prenatal Hypoxia

The cause of fetal growth restriction is the dysfunction of the placental bed, which impacts the diffusion of gases and may lead to chronic hypoxia. Complications caused by low oxygen concentration are observed postnatally and during the course of childhood. The most severe manifestation of oxygen shortage is hypoxic-ischemic encephalopathy (HIE), which is a significant cause of perinatal mortality and neural disability in newborns. Despite initiating the potential for brain cell recovery after perinatal asphyxia, neurologic sequelae proceed during the early postnatal period due to progressive failure of the cerebral oxidative metabolism [12,13]. In addition, as the oxygen concentration decreases, the fetal metabolism switches to anaerobic glycolysis, which causes the accumulation of lactic acid and eventually leads to electrolyte imbalance. This process results in cell edema, depolarization, the disruption of organelles, and finally apoptosis. Moreover, Rainaldi and Perlman et al. showed the relationship between severe fetal acidemia (pH of less than 7.00) and seizures, which are common symptoms that reflect brain damage [14].

To standardize the assessment of HIE, the Sarnat staging for encephalopathy was introduced (Figure 1).

Sarnat staging distinguishes neonates into three groups based on their clinical state. Symptoms such as hyper-alertness, overreactive stretch reflexes, mydriasis, or tachycardia classify neonates into stage 1 of Sarnat staging. Newborns in stage 2 are lethargic and have weak stretch reflexes and muscle tone; seizures are commonly observed. Most severe cases that have comas, absent primate reflexes, and absent activity as symptoms are described as stage 3. This staging is currently used as a prognostic tool and as therapeutic hypothermia (TH) criteria. According to the authors, infants that presented the symptoms of stage 2 (but not stage 3) for less than five days had no complications in further development; an elongation of that period to more than seven days or to failure of the EEG to revert to normal was related to later neurologic impairment or death [15]. The study of Giannopoulou et al. revealed that perinatal hypoxia might be reflected in later periods as it predisposes one to neurological and cognitive deficits in adulthood [16]. Similarly, Modabbernia et al. underline that hypoxic conditions are among the risk factors for autism spectrum disorder [17].

## 4. Assessment of Brain Damage during Neurological Development in the Growth-Restricted Human Fetus

A spectrum of alterations in morphological neurostructural and functional brain development reflects the severity of FGR and uteroplacental dysfunction during pregnancy. Due to chronic placental insufficiency (CPI), the transfer of oxygenated blood in the fetal organism is limited. Therefore, the fetus develops hemodynamical adaptations that promote blood redistribution to vital organs, such as the brain, heart, and adrenals. However, this mechanism, known as brain sparing, does not ensure proper intrauterine growth [18]. Furthermore, FGR increases the risk of preterm birth, enhancing the degree of possible neurological impairment in premature infants [19].

Neurological complications in pregnancies affected by FGR include changes in brain volume development, morphology, and trajectory. The imaging and assessment of these features pre- and postnatally can be obtained through the use of ultrasound scans with biometry, Doppler flow and extensive neurosonography, or magnetic resonance imaging techniques. Three-dimensional measurements of fetal brains in ultrasound scans have shown that total brain volume, head circumference (HC), and biparietal diameter (BPD) were smaller in a group of FGR fetuses than in a group of appropriate for gestational age (AGA) fetuses. This pattern corresponded to significantly lower mid-cerebral artery (MCA) Doppler indices (PI and RI) in the FGR group, indicating brain sparing [20,21]. The assessment of brain volume for the FGR and AGA fetuses using quantitative 3D MRI conducted by Adescavage et al. presented a similar relationship.

Additionally, measurements of both the cerebrum and cerebellum volume were decreased in fetuses with FGR, especially in subgroups with abnormal fetal umbilical artery Doppler studies. In contrast, there was no statistical difference in the brainstem volume between FGR (with or without abnormal Doppler flows) and AGA groups [22]. Postnatally, a comparison of the gray matter development of FGR and AGA infants at 12 months was presented in the work of Padilla et al. Using voxel-based morphometry, the researchers observed a reduced volume in the amygdala, basal ganglia, thalami, insula, bilateral angular gyri, left parietal and occipital lobes, and the right perirolandic region [23]. These changes may implicate both movement and mental disorders in the future. Hippocampus development is the least genetically regulated brain region; therefore, it is susceptible to external factors such as chronic hypoxia during pregnancy. Infants with FGR had a reduced volume in the hippocampal region at birth and during follow-up at the age of 2 years compared with the control group. A smaller volume of the hippocampus is correlated with altered attention and poor performance in motor skills and motor activities [21]. Changes in the cortical development between small fetuses and AGA may be visualized using neurosonography. As presented in the work of Paules et al., both the SGA and FGR groups showed differences in insular depth and corpus callosum length. The FGR individuals were characterized as having the shortest corpus callosum and the most reduced Sylvian fissure depth. These changes were assessed as statistically significant and supported the role of neurosonography in picturing subtle changes in brain damage [24].

Limited maturation of gray matter inhibits proper synaptogenesis and dendritic branching and therefore affects intraneuronal transmission (Figure 2). Furthermore, Dudink et al. suggested that trajectory complications involve the whole brain and interhemispheric connections [20]. When considering the white matter of the brain, FGR induces the degeneration of oligodendrocytes that are particularly vulnerable to oxidative stress. Another pathology is periventricular leukomalacia (PVL); it is a white matter lesion, which not only occurs in fetuses with FGR due to hypoxia but also occurs due to ischemia and coagulation necrosis. PVL consists of one or more small (a few millimeters), rounded periventricular cysts. They emerge superiorly and laterally from the border of the lateral ventricle in the watershed zones of five arteries (anterior cerebral artery, MCA, anterior, Heubner, and lenticulostriate choroidal arteries), and they may exist on one or both sides. Hyperechogenicity of parenchyma may be seen around the cysts; they may also be found posteriorly to the lateral ventricle. Moreover, the consequence of hypoxia or hemorrhage is the formation of periventricular pseudocysts known as subependymal or germinolytic cysts, which are commonly visualized during the third trimester. The cysts on T2-weighted and T1-weighted MRI images are hyperintense and hypointense, respectively. If hemorrhaging is the cause, a susceptibility artefact on the gradient echo sequence is seen. MRI imaging may miss lesions that are less than 5 mm. Neurosonography helps to detect these lacerations [25].

## 5. Animal Models of FGR

For a better understanding of human health and disease, animal models are essential. When choosing the most appropriate model for FGR testing, many aspects should be considered, such as human organogenesis, the development and structure of the placenta, the litter size, the length of gestation, and the possibility of genetic modifications. The most popular animal species used in research are mice and rats (79%) and sheep (16%). The remainder includes guinea pigs, rabbits, chickens, pigs, nonhuman primates, and other species [26].

Mice and rats have many advantages when used in FGR research. The length of gestation (19–21 days in mice and 21–23 days in rats) and the small size of the animals with many pups in the litter make them an inexpensive model compared with other species, and they can be quickly analyzed. Mice and rats, like human infants, show altricial brain development. In the work of Reid et al., researchers used the bilateral ligation of uterine arteries in embryonic rats to mimic placental insufficiency. After eight weeks, the oligodendrocyte maturation and myelination process in the periventricular region were delayed. These changes were related to behavioral deficiencies in the further observation of the animals [27].

Humans, as well as rats and mice, share many similarities with respect to hemochorial placentas. Due to the limited cellular barrier, the placentas act efficiently and deliver nutrients between the mother and the fetus. However, the structure and development of mouse and rat placentas are not identical to that of humans. Animal placentas are hemotrichoral; there are one cytotrophoblasts layer and two syncytiotrophoblast layers that separate the maternal and fetal circulation. On the other hand, the human placenta in late gestation is hemomonochorial with one syncytiotrophoblast layer. Moreover, rats and mice show relatively shallow uterine wall invasions compared with humans [28]. Additionally, the exchange region develops as a labyrinth rather than in cotyledons as it does in humans [29].

Despite these differences, numerous studies using transgenic mice have been conducted due to the well-studied genome and the possibility of genetic modification. Knockouts of genes that are involved in various mechanisms are used in conducting studies. The most common gene modifications involve vascular endothelial growth factor (VEGF), insulin-like growth factors (IGF-1 and 2), and nitric oxide synthase (eNOS). A systemic knockout of the gene encoding VEGF results in abnormal blood vessel formation and defective extraembryonic vascularization, which leads to disturbances in the mother–fetus transport of nutrients and oxygen and develops a lethal embryonic phenotype [30].

Homozygote-specific IGF-1 knockout mice showed a significant decrease in placental efficiency, causing FGR [31]. In the placental specific IGF-2 knockout mouse model, a product of this gene, which is only expressed in the labyrinthine placental trophoblast, is deleted. This results in a growth restriction in 96% of fetuses by embryonic day 16. Pups exhibit postnatal catch-up growth FGR [32]. Increasing evidence has shown that the targeted administration of IGF-1 or IGF-2 effectively affects fetal growth in FGR models [33]. A systemic knockout of the eNOS gene, which encodes the converting enzyme of arginine to nitric acid (NO), results in hypertension, impaired uterine artery function, and depletion of the placental system. The amino acid transporter activity level correlates with the severity of FGR in humans [34,35,36].

Rats represent a sufficiently sizeable animal to perform complex surgical procedures on, including in FGR model development. The most popular rat strains used in FGR research are the Sprague Dawley and Wistar strains [37]. Studies have included uterine artery ligation [38], and the occlusion showed the limited occurrence of FGR with a high level of fetal mortality/loss [39]. The same results have been achieved by administering dexamethasone and L-NAME [40,41]. Furthermore, diet-induced FGR (nutritional or protein restriction) has not been shown to exhibit as elevated results as the transgenic model [42].

The most human-like organogenesis and placenta structure can be found in other rodents that are used in research, such as guinea pigs. Guinea pigs have a hemochorial placenta, which contains a single layer of trophoblast [43], and its development is associated with deeper trophoblast invasion compared with mice and rats [44]. Unlike humans, guinea pig organ development is precocial. Due to the gestation period (65–70 days), the animal model is suitable for performing multiple measurements, but this results in higher experimental costs than mice and rats. The FGR model in guinea pigs was achieved using surgical methods such as artery ligation and nutrient restriction [45]. Some studies indicate that FGR affects postnatal brain development and function. FGR results in reduced brain size and increased apoptosis of the white matter and hippocampus [46], as well as reduced granule cell proliferation, Purkinje cell expansion, and impaired cerebellar development [47].

In turn, sheep represent an alternative animal model that is used in the development of FGR. Fetal development milestones share similarities to humans, including gestation length (144–151 days) and litter size (single or twin pregnancy) [48]. However, sheep develop the synepitheliochorial placenta (six layers of tissue between maternal and fetal blood) compared with the syncytiotrophoblast layer in late gestation in the hemomonochorial human placenta. In addition, multiple placental sites consist of trophoblast-rich cotyledon tissue that is in contact with a caruncle [45]. The maternal–fetal anatomical arrangement of blood vessels is crucial to placental efficiency. However, crosscurrent flows in sheep are comparable to a human countercurrent arrangement. Modeling of the sheep FGR includes surgical techniques (carunculectomy [49,50] and umbilical artery ligation [51]), diet-induced FGR (nutrient restriction or overfed [52,53]), and maternal hyperthermia [54,55] or hypoxia [56].

## 6. Known Molecular Pathways of Abnormal Brain Development in a Hypoxic Fetus

Due to CPI, the hypoxic and acidotic condition affects brain function in the puerperium and even increases the risk of neurodegenerative disorders later in life. The effects of hypoxia, in the form of cerebral palsy and cognitive and behavioral deficits in the postpartum period, correlate with structural changes in the brain. The areas responsible for memory and learning are particularly vulnerable to hypoxia. Morphological changes were found in the sensorimotor cortex and hippocampus. In recent years, the presence of abnormalities of mature neurons in the cerebellar external granular layer (EGL) caused by chronic prenatal hypoxia has been shown in the CPI animal model of the guinea pig [57].

Apart from the morphological changes, as a result of hypoxia, the adaptive potential of the brain, including the ability to create connections between cells and properly propagate neuronal stimuli, all collapse. The above abnormalities are accompanied by a change in gene expression, which results in disturbances at the stage of the post-translational modification of proteins that are involved in normal brain function. The enzymes that are most exposed to prenatal hypoxia are acetyl and butyryl cholinesterases (AChE and BchE) [58], the amyloid precursor protein (APP) [59], and peptidases that are involved in catabolism neuropeptides, including amyloid-β peptide (Aβ). As a result, a progressive deficit in Aβ clearance was observed, which contributed to the progression of neurodegeneration, thus predisposing individuals to the development of Alzheimer’s disease pathology later in life [60,61]. It was proven that prenatal hypoxia increased the level of membrane-bound APP protein in the sensorimotor cortex and at the same time decreased the level of its soluble forms (sAPP), which have a protective neurotogenic effect [62]. Animal studies have confirmed that prenatal hypoxia is involved in the pathogenesis of Alzheimer’s disease [63]. For example, it has been shown that a hypoxic state in the prenatal period induces a change in the activity of α-secretase regulating sAPPα release, which prevents the formation of Aβ [64]. At the same time, a deficiency in amyloid-degrading enzymes in the brain, i.e., neprilysin, endothelin converting enzyme, and angiotensin converting enzyme, was observed [65]. The observation made by Vasilev et al. showed elevated levels of retinol transporter transthyretin (TTR) in the choroid plexus of prenatal hypoxia puppies [66]. TTR is known to play an essential role in the development of the fetal brain; it aims to protect the brain from Aβ accumulation. Note that the deformation of this protein due to hypoxia potentially results in TTR amyloidosis [67]. In turn, changes in the activity of enzymes such as AChE and BChE may influence immune and stress responses [68].

Studies conducted on rats have revealed the altered expression and activity of peptidases, e.g., neprilysin and endothelin converting enzymes [65], which altered the adenylate cyclase pathway in the striatum and was responsible for learning deficits [69]. Tong et al. showed an increased activity of matrix metallopeptidases (MMPs) and a decreased expression of a tissue metalloproteinase inhibitor (TIMP) in the brain of prenatally hypoxic neonatal rats [70]. The observed biochemical changes may negatively affect the structure of neural circuits during brain development. Detection of biological biomarkers that are sensitive to neurological injury may additionally represent the grade of brain sparing in FGR. One of the markers of neuronal damage includes the protein S100β, which is used as a prognostic marker in circulatory arrest or stroke. Elevated levels of S100β in patients with head trauma correspond to the severity of the injury and possible therapeutic outcome [71,72]. In the work of Swiss et al., the concentration of S100β protein was assessed in maternal plasma in SGA pregnancies, FGR pregnancies, and FGR pregnancies with brain sparing in ultrasound scans. The S100β protein concentration was found to be the highest in the last group and corresponded to increased levels of the sFLT-1/PIGF ratio [73]. In recent years, several excellent reviews of the role of prenatal hypoxia in brain development have been published. The consequences of hypoxia have been discussed for the puerperium and subsequent years after due to the risk of neurodegenerative disorders [74,75].

## 7. Gaps in Understanding the Detrimental Effect of Hypoxia on the Fetal Brain

Two primary causes of placental prenatal hypoxia have been identified: environmental and placental. Hypoxia results in the overproduction of ROS and nitrogen (RNS) free radicals, inflammation, and apoptosis. Most authors have also considered changes in gene expression, glutamatergic excitatory damage, and the role of NMDA receptors. In animal models, structural changes due to ICP have been confirmed in different brain structures, especially in the striatum and hippocampus. Neuroimaging studies have revealed the localization of brain changes due to hypoxia. Lesions were identified in cortical structures and subcortical gray matter, including the basal ganglia, thalamus, and hippocampus. McClendon highlighted a significant loss in neurons in the cerebral cortex [76]. In his review, Wang distinguished several of the most important mechanisms that are responsible for the occurrence of neurological disorders. Among others, abnormal modifications of epigenetics, endocrine axis disorders, oxidative damage, and mitochondrial dysfunction were observed [77].

Recently, most attention has been paid to research on epigenetic mechanisms in fetal brains that are activated by prenatal hypoxia [78,79]. The mechanisms underlying the effects of chronic fetal hypoxia and FGR on epigenetic programming of fetal brain endothelial cells still require investigation [80].

Another area of modern research is disturbed integrity in neurovascular units (NVUs), which is ensured by, among other factors, the blood–brain barrier (BBB), which is built from a monolayer of brain endothelial cells [81]. A cerebrovascular tree is a multicellular unit where glial cells (i.e., astrocytes, microglia), pericytes, and neurons closely interact. The BBB breakdown appears to be the principal factor of cerebrovascular diseases [82]. The NVU and perinatal programming are currently a subject of consideration. However, this area of research, which includes the influence of prenatal hypoxia on changes in the expression of proteins that are responsible for the integrity of the cerebrovascular network, still needs to be fully understood. Hypoxia-induced BBB permeability studies seem to play a key role in understanding these mechanisms [83]. The role of the glucocorticoid receptor (GR) isoform in adapting to stressful conditions, including intrauterine hypoxia, also requires subsequent research. Clifton et al. demonstrated different types of GR proteins that are responsible for various types of placental responsiveness, which are further related to pregnancy complications and fetal sex [84].

## 8. Possible Treatment Modalities for the Hypoxic Brain

The current approach in the treatment of HIE in neonates is based on therapeutic hypothermia (TH). The procedure is also applied in intraventricular hemorrhage and periventricular leukomalacia, especially in preterm neonates [85]. The mechanism of TH involves a reduction in extracellular excitatory neurotransmitters such as dopamine and glutamate. In normothermic conditions, the concentration of these substances increases due to cerebral asphyxia and eventually causes hyperexcitability. This metabolic pathway initiates an excitotoxic cascade. Additionally, hypothermia promotes the production of brain-derived neurotrophic factor (BDNF), which is one of the enzymes used in preserving neuronal integrity [86]. Finally, Chakkarapani et al. showed that the application of TH delays the inflammatory response of the organism due to perinatal asphyxia. The increase in CPR, WBC, and platelets was lower in the group with TH than in the control group [87].

In 2005, the first multicenter randomized study, the CoolCap trial, revealed that selective head cooling for 72 h improved survival without neurodevelopmental impairment in mixed populations of infants, and this approach did not affect the mortality rate [88]. Additionally, in neonates that had the most severe changes in their aEEG, no beneficial effect of TH was observed in the 18-month follow-up. On the contrary, a recent meta-analysis including seven randomized trials of 1214 newborns provided optimistic conclusions: TH reduced the risk of death and major neurodevelopmental disability [89]. Recently, researchers have investigated multiple novel therapies to prevent perinatal hypoxia in prenatal and antenatal periods. The enrolled medical trials have concentrated on different drugs, such as magnesium sulphate (MgSO_4_) [90], erythropoietin (Epo) and darpoetin, allopurinol, indomethacin, topiramate (TPM), and melatonin and xenon [85]. The goal of the implementation of MgSO_4_ is for fetal neuroprotection in pregnancies that are at risk of preterm birth. Similarly, MgSO_4_ inhibits the excitotoxic cascade due to the inactivation of NMDA receptors, which is followed by increased intracellular calcium concentration reduction. The mechanism also includes anti-inflammatory properties such as decreased production of free radicals and cytokines after hypoxia [91]. Consequently, MgSO_4_ prevents cell death due to perinatal hypoxia. In clinical management, the antenatal and postnatal application of magnesium sulphate in pregnancies with a risk of preterm birth has resulted in a smaller ratio of both cerebral palsy and substantial gross motor dysfunction cases [92]. In the acute phase of hypoxia, the antioxidant properties of allopurinol and melatonin are beneficial. Regarding studies using rodents, allopurinol decreased the edema of the brain and sub-acute brain atrophy; however, it did not reduce the concentration of S100β in the cord blood in fetal asphyxia [85,93].

Researchers investigated the benefits of introducing the Mediterranean diet and mindfulness-based stress reduction on FGR. The main objective of the Improving Mothers for a Better Prenatal Care Trial Barcelona (IMPACT BCN) protocol is to reduce the prevalence of FGR and adverse perinatal outcomes through the implementation of changing the dietary patterns (including the increased intake of unsaturated fats such as olive oil), meditation, and performing prenatal yoga positions. Moreover, the research will examine the neurodevelopmental and cardiological profile of the children in a 2-year follow-up. The results of this first randomized study are still under development [94].

## 9. Conclusions

Perinatal hypoxia continues to be a primary cause of neurodevelopmental impairment and subsequent motor and cognitive dysfunctions. As most asphyxia-induced brain injuries occur during pregnancy, careful monitoring of fetal wellbeing is substantial. Undoubtedly, an ultrasound scan is essential for a perinatal checkup of the fetus. Ultrasound scans allow for the assessment of Doppler flows with precise measurement of the fetal brain and help to determine the possibility of the fetuses undergoing hypoxia in pregnancies with complications such as FGR. Novel diagnostic procedures, such as neurosonography and antenatal MRI, are used to identify which structures of the brain have been particularly affected. In neurosonography, a typical image of brain sparing in FGR fetuses includes changes in the cortical development, short corpus callosum, and most reduced Sylvian fissure depth. MRI technology indicates smaller brain volume with the cerebrum and cerebellum being significantly decreased in FGR cases, followed by abnormal Doppler flows in US scans. Limited maturation of gray and white matter may explain impairments in interhemispheric trajectories. Molecular changes resulting from perinatal asphyxia led to post-translational modification of specific enzymes such as acetyl and butyryl cholinesterases. Moreover, in hypoxia, the level of membrane-bound APP protein in the sensorimotor cortex increased, while the soluble form, which is noted in neuroprotective properties, decreased. Authors have also concentrated on the influence of matrix metallopeptidases and S100β protein concentration. For a better understanding of perinatal hypoxia and in the search of innovative therapies, experimental trials have concentrated on using animal models; however, the differences in placental structure continue to be a significant factor. In contrast to humans, other rodent species used in FGR animal models include guinea pigs, whose organ development is precocial. Moreover, novel studies including the application of surgical techniques have been conducted on sheep models. Both antenatal and postnatal modern management of perinatal hypoxia is limited, and both require further research.

## Figures and Tables

**Figure 1 ijms-24-10401-f001:**
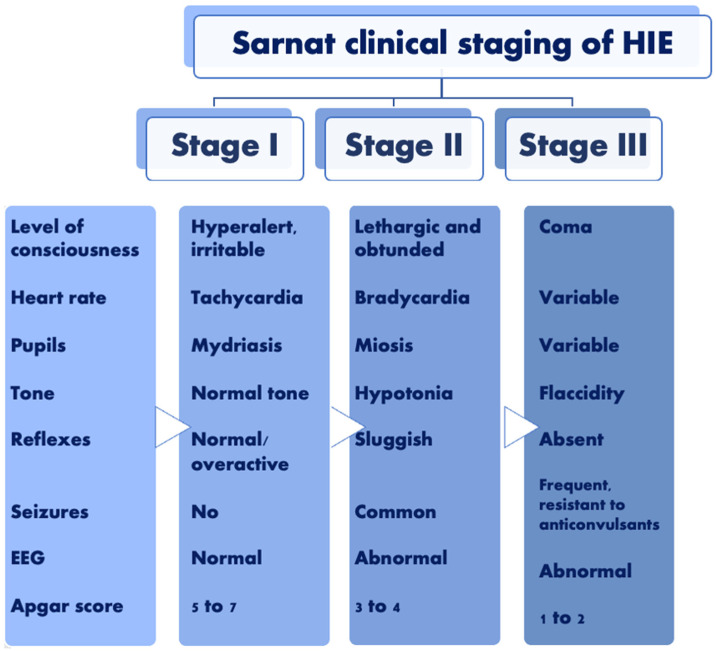
Sarnat clinical staging of HIE based on [15].

**Figure 2 ijms-24-10401-f002:**
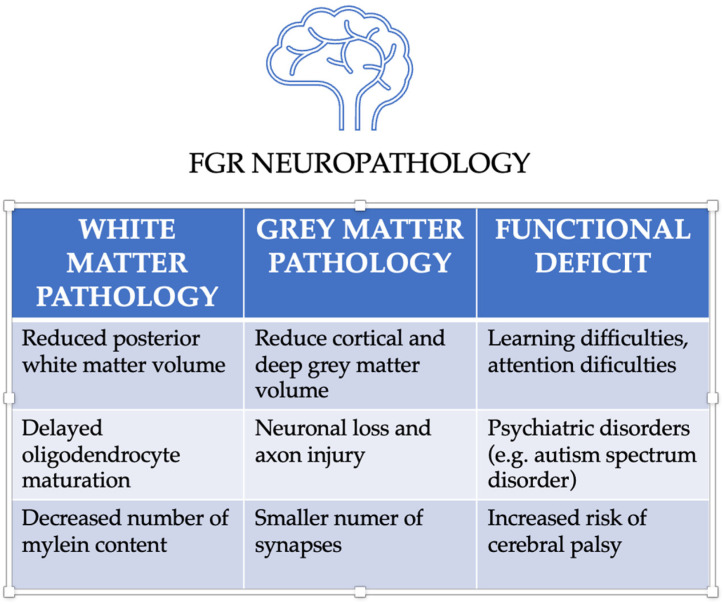
FGR neuropathology. White matter and great matter pathology with the functional deficits indicated based on [20,22].

## Data Availability

No new data were created.

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
