# Peer review of "Molecular Pathways of Altered Brain Development in Fetuses Exposed to Hypoxia"

_ijms, 2023, doi:10.3390/ijms241210401_

Round 1
Reviewer 1 Report
This review present the current knowledge on brain development resulting from perinatal hypoxia, including causes, symptoms and means of predicting the degree of brain damage.It also discusses the specificity of brain development in the growth restricted fetus and and how it is replicated and studied in animal models.
Finally, it aims at identifying the pathways of abnormal brain development in the aspect of potential treatment.
In conclusion, this manuscript provides a useful and comprehensive review of altered brain development in fetuses exposed to hypoxia.
This is nice review on altered brain development in fetus exposed to hypoxia.
It reads well and is well documented
Author Response
Review 1
This review present the current knowledge on brain development resulting from perinatal hypoxia, including causes, symptoms and means of predicting the degree of brain damage.
It also discusses the specificity of brain development in the growth restricted fetus and and how it is replicated and studied in animal models.
Finally, it aims at identifying the pathways of abnormal brain development in the aspect of potential treatment.
In conclusion, this manuscript provides a useful and comprehensive review of altered brain development in fetuses exposed to hypoxia.
This is nice review on altered brain development in fetus exposed to hypoxia.
It reads well and is well documented
Answer: Thank you for your kind review.

Reviewer 2 Report
Line 25: Please consider: has increased primarily due to
Line 49: please consider replacing the dash with a colon: two main independent mechanisms: Placental and environmental hypoxia.
Line 52: The word diabetes appears to be missing: As an example, gestational diabetes mellitus…
Line 85: Please consider replacing “on the contrary” with “on the other hand”.
Line 90: Consider deleting the word: “Inadequate”.
Line 117: Please consider changing: “Might reflect” to: “might be reflected”.
Line 143: Please consider changing: “On the contrary” to “in contrast”.
Line 146. Please consider beginning in mucins at the word using.: Using voxel based morphometry..
Line 154: please add: motor skills or motor activities
Line 174: Please consider changing to posterior to the lateral ventricle.
Line 217: Please consider defining or deleting “FRG”
Line 225: Sprague-Dawley
Line 226: In: “The limited” please considered leading the word “the”.
Line 228: Please correct for logic: 1st part of the paragraph is about vascular models. The text then states that dietary models are not as effective as transgenic models but dietary models were never previously discussed and furthermore it is not clear which genetic models are being referred to.
Line 230: Please correct the 1st sentence for clarity and logic. It is not clear what the following sentence means: “The most similar to human organogenesis and placenta structure have other rodents used in research”.
Line 241: Please considered deleting the words: “in turn”.
Content
Line 84: The phrase” Due to their pro-apoptotic effect, HIFs promote tumor progression” requires clarification because usually apoptosis is thought of as limiting tumor progression.
Line 246: please change tropho0blast to trophoblast
Lines 248 and 249: Please clarify further the concepts of cross current and counter current or alternatively it may be reasonable to leave this part out
Line 264: Please consider changing the phrase: “all collapse” to: “all are affected”.
Line 306: Please consider changing the beginning of the sentence to: “Consequences of hypoxia are”
Line 332: Please further explain: “Placental responsiveness”.
Line 338: Please clarify the application of therapeutic hypothermia to periventricular leukomalacia. My understanding is that administration of therapeutic hypothermia to an infant with hypoxic ischemic encephalopathy reduces the risk of subsequent development of periventricular leukomalacia; however, it is difficult to envision using cooling for PVL itself, because PVL usually develops outside of the neonatal setting.
Line 349: Please consider changing to: “In a mixed population of infants but did not affect the mortality rate”. Please also explain what mixed population means.
Line 359: Should it not be: “activation of NMDA receptor” ?
Line 386: Please change the syntax of the sentence beginning with: “the authors
Line 25: Please consider: has increased primarily due to
Line 49: please consider replacing the dash with a colon: two main independent mechanisms: Placental and environmental hypoxia.
Line 52: The word diabetes appears to be missing: As an example, gestational diabetes mellitus…
Line 85: Please consider replacing “on the contrary” with “on the other hand”.
Line 90: Consider deleting the word: “Inadequate”.
Line 117: Please consider changing: “Might reflect” to: “might be reflected”.
Line 143: Please consider changing: “On the contrary” to “in contrast”.
Line 146. Please consider beginning in mucins at the word using.: Using voxel based morphometry..
Line 154: please add: motor skills or motor activities
Line 174: Please consider changing to posterior to the lateral ventricle.
Line 217: Please consider defining or deleting “FRG”
Line 225: Sprague-Dawley
Line 226: In: “The limited” please considered leading the word “the”.
Line 228: Please correct for logic: 1st part of the paragraph is about vascular models. The text then states that dietary models are not as effective as transgenic models but dietary models were never previously discussed and furthermore it is not clear which genetic models are being referred to.
Line 230: Please correct the 1st sentence for clarity and logic. It is not clear what the following sentence means: “The most similar to human organogenesis and placenta structure have other rodents used in research”.
Line 241: Please considered deleting the words: “in turn”.
Line 246: please change tropho0blast to trophoblast
Lines 248 and 249: Please clarify further the concepts of cross current and counter current or alternatively it may be reasonable to leave this part out
Line 264: Please consider changing the phrase: “all collapse” to: “all are affected”.
Line 306: Please consider changing the beginning of the sentence to: “Consequences of hypoxia are”
Line 332: Please further explain: “Placental responsiveness”.
Line 338: Please clarify the application of therapeutic hypothermia to periventricular leukomalacia. My understanding is that administration of therapeutic hypothermia to an infant with hypoxic ischemic encephalopathy reduces the risk of subsequent development of periventricular leukomalacia; however, it is difficult to envision using cooling for PVL itself, because PVL usually develops outside of the neonatal setting.
Line 349: Please consider changing to: “In a mixed population of infants but did not affect the mortality rate”. Please also explain what mixed population means.
Line 359: Should it not be: “activation of NMDA receptor” ?
Line 386: Please change the syntax of the sentence beginning with: “the authors”
Author Response
Review 2 Comments and Suggestions for Authors
Line 25: Please consider: has increased primarily due to
Line 49: please consider replacing the dash with a colon: two main independent mechanisms: Placental and environmental hypoxia.
Line 52: The word diabetes appears to be missing: As an example, gestational diabetes mellitus…
Line 85: Please consider replacing “on the contrary” with “on the other hand”.
Line 90: Consider deleting the word: “Inadequate”.
Line 117: Please consider changing: “Might reflect” to: “might be reflected”.
Line 143: Please consider changing: “On the contrary” to “in contrast”.
Line 146. Please consider beginning in mucins at the word using.: Using voxel based morphometry..
Line 154: please add: motor skills or motor activities
Line 174: Please consider changing to posterior to the lateral ventricle – I do not understand this comment
Line 217: Please consider defining or deleting “FRG” – do you mean FGR – it was defined in the introduction
Line 225: Sprague-Dawley
Line 226: In: “The limited” please considered leading the word “the”.
Line 228: Please correct for logic: 1st part of the paragraph is about vascular models. The text then states that dietary models are not as effective as transgenic models but dietary models were never previously discussed and furthermore it is not clear which genetic models are being referred to.
Line 230: Please correct the 1st sentence for clarity and logic. It is not clear what the following sentence means: “The most similar to human organogenesis and placenta structure have other rodents used in research”.
Line 241: Please considered deleting the words: “in turn”. – not found in the text
Answer: Thank you for your review. All suggestions are included in the revised version.

Reviewer 3 Report
The authors have done a great job in putting together a concise review stating the effects of hypoxia on fetal neurodevelopment which I find extremely important. If the authors can elaborate and provide some more information on the available therapeutic implementations available be it drugs or surgery(the authors have already mentioned some in the last section), to treat neonates born with FGR and if there is any supplements or dietary restrictions that a mother can follow during pregnancy that might improve hypoxic conditions then that will great addition to the review. Is it possible to add more figures of hypoxic versus control normal placentae?
Please correct the following typos:
In Fig2: please correct the spelling of 'myelin'.
In line 199: please correct the spelling of 'haemotrichorial'.
Please rewrite line 230, it's grammatically incorrect.
In line 231: please correct spelling of 'haemochorial' or 'hemochorial' and not hemochoric.
Author Response
Review 3
Comments and Suggestions for Authors
The authors have done a great job in putting together a concise review stating the effects of hypoxia on fetal neurodevelopment which I find extremely important. If the authors can elaborate and provide some more information on the available therapeutic implementations available be it drugs or surgery (the authors have already mentioned some in the last section), to treat neonates born with FGR and if there is any supplements or dietary restrictions that a mother can follow during pregnancy that might improve hypoxic conditions then that will great addition to the review. –
Answer: Currently there are very few prenatal intervention that can stop the hypoxic process. We believe this elaboration goes a bit beyond the scope of this review but we have made some revisins of that paragraph. That is why there is only a short mention of possibilities. This is a great topic for a systematic review of current literature. Perhaps it will be the topic of our follow up research. Thank you for this suggestion.
Is it possible to add more figures of hypoxic versus control normal placentae?
Answer: Actually originally we added ultrasound images of normal an hypoxic fetal brains but the journal did not allow to print them as they are original work.
Comments on the Quality of English Language
Please correct the following typos:
In Fig2: please correct the spelling of 'myelin'.
In line 199: please correct the spelling of 'haemotrichorial'. –
Please rewrite line 230, it's grammatically incorrect.
In line 231: please correct spelling of 'haemochorial' or 'hemochorial' and not hemochoric.
Answer: The above suggestions were included in the final version of the manuscript.